# TRAPing Ghrelin-Activated Circuits: A Novel Tool to Identify, Target and Control Hormone-Responsive Populations in TRAP2 Mice

**DOI:** 10.3390/ijms23010559

**Published:** 2022-01-05

**Authors:** Iris Stoltenborg, Fiona Peris-Sampedro, Erik Schéle, Marie V. Le May, Roger A. H. Adan, Suzanne L. Dickson

**Affiliations:** 1Department of Physiology/Endocrine, Institute of Neuroscience and Physiology, The Sahlgrenska Academy at the University of Gothenburg, 405 30 Gothenburg, Sweden; iris.stoltenborg@gu.se (I.S.); fiona.peris.sampedro@gu.se (F.P.-S.); erik.schele@medic.gu.se (E.S.); marie.lemay@neuro.gu.se (M.V.L.M.); 2Department of Translational Neuroscience, Brain Center Rudolf Magnus, University Medical Center Utrecht and Utrecht University, 3584 CJ Utrecht, The Netherlands; r.a.h.adan@umcutrecht.nl

**Keywords:** ghrelin, arcuate nucleus, GHSR, AgRP, food intake, food choice, chemogenetics, DREADD, Fos-TRAP

## Abstract

The availability of Cre-based mouse lines for visualizing and targeting populations of hormone-sensitive cells has helped identify the neural circuitry driving hormone effects. However, these mice have limitations and may not even be available. For instance, the development of the first ghrelin receptor *(Ghsr)-IRES-Cre* model paved the way for using the Cre-lox system to identify and selectively manipulate ghrelin-responsive populations. The insertion of the *IRES-Cre* cassette, however, interfered with *Ghsr* expression, resulting in defective GHSR signaling and a pronounced phenotype in the homozygotes. As an alternative strategy to target ghrelin-responsive cells, we hereby utilize *TRAP2* (targeted recombination in active populations) mice in which it is possible to gain genetic access to ghrelin-activated populations. In *TRAP2* mice crossed with a reporter strain, we visualized ghrelin-activated cells and found, as expected, much activation in the arcuate nucleus (Arc). We then stimulated this population using a chemogenetic approach and found that this was sufficient to induce an orexigenic response of similar magnitude to that induced by peripheral ghrelin injection. The stimulation of this population also impacted food choice. Thus, the TRAPing of hormone-activated neurons (here exemplified by ghrelin-activated pathways) provides a complimentary/alternative technique to visualize, access and control discrete pathways, linking hormone action to circuit function.

## 1. Introduction

The brain ghrelin signaling system comprises neuronal networks that are targeted by the stomach-derived hormone, ghrelin [1], whose primary role is to ensure that we seek out and consume food [2,3,4]. Other ligands targeting this system include synthetic growth hormone secretagogues (GHS) [5] and also liver-enriched antimicrobial peptide 2 (commonly referred to as LEAP2), an endogenous inverse agonist [6,7]. These hormones/compounds target the populations of neurons expressing the ghrelin receptor (the GHS receptor, GHSR) and downstream pathways. It remains a challenge to know which populations of ghrelin-responsive neurons drive specific food-linked behaviors, including food intake [8,9,10], food choice [11,12], food anticipation [13,14], food motivation [2,15,16] and food reward [17]. Unraveling the role of the specific populations of ghrelin-responsive neurons has often involved the site-specific injections of ghrelin or GHSR ligands, which induce an orexigenic response at most brain areas targeted, including the arcuate nucleus (Arc) and many others [16,17,18,19,20,21,22]. Such approaches invariably raise questions, however, about the spread of the hormone from the injection site and do not provide information about the precise populations of cells activated by ghrelin.

The emergence of chemogenetic approaches (including designer receptors exclusively activated by designer drugs, or DREADD) opened up new possibilities to explore the function of specific populations of neurons within a network. In the ghrelin field, the group of Jeffrey Zigman developed a novel mouse strain, the *Ghsr*-*IRES*-*Cre* mouse, in which *Cre* recombinase (Cre) expression is controlled by the endogenous *Ghsr* promotor [23]. The presence of Cre in the GHSR-expressing neurons provides genetic access to them, enabling them to be visualized and chemogenetically controlled in order to determine their function. In this article, they demonstrated that the chemogenetic inhibition of GHSR-expressing cells in the mediobasal hypothalamus (which includes the Arc) suppressed the orexigenic response to ghrelin and also that the chemogenetic stimulation of these cells caused a feeding response. We have recently shown, however, that inserting an *IRES-Cre* cassette into the 3′-untranslated region of the *Ghsr* gene interferes with *Ghsr* expression, since *Ghsr* expression was decreased in heterozygotes and absent in homozygotes [24]. Moreover, there appears to be both intra- and inter-individual variability in terms of the expression of *Ghsr* in *Ghsr-IRES-Cre* mice crossed to different reporter strains [23], findings that we have also observed (unpublished). These limitations make *Ghsr-IRES-Cre* mice less than ideal for teasing apart the function of different populations of GHSR-expressing neurons.

In the present study, we explore an alternative strategy to chemogenetically control (and hence, explore the function of) populations of ghrelin-responsive neurons, here targeted because they express *Fos* when ghrelin-activated rather than because they are GHSR-expressing. In our proof-of-concept study, we focus in particular on the ghrelin-activated population in the Arc, an area strongly activated by ghrelin and GHS [25,26,27]. To gain genetic access (via Cre) to neurons expressing *Fos* in vivo in mice, we utilize the Fos-TRAP (targeted recombination in active populations) system [28]. Specifically, we use *TRAP2* mice [29], in which *Fos*-expressing cells co-express inducible Cre (known as CreER^T2^) that enters the nucleus upon binding to 4-hydroxytamoxifen (4-OHT) leading to recombination. Thus, when 4-OHT is present, it defines a time-window during which ghrelin-activated neurons become “TRAPed” to express a target gene (allowing those cells to be visualized and/or controlled). To validate the *TRAP2* mouse model as an alternative tool for exploring pathways targeted by ghrelin, ghrelin-activated neurons in the Arc were chemogenetically stimulated prior to assessing feeding behavior (both food intake and food choice) and *Fos* expression. To demonstrate the engagement of the orexigenic systems, we further explored whether the cells activated by a DREADD agonist (clozapine-N-oxide, CNO) colocalized with agouti-related peptide (AgRP) using RNAscope.

## 2. Results

### 2.1. Chemogenetic Stimulation of Ghrelin-Activated Cells in the Arc Leads to Increased Food Intake in TRAP2 Mice

To explore the orexigenic capacity of TRAPed cells in the Arc (ghrelin-activated versus background-activated), we measured food intake after the chemogenetic activation of these populations. Food intake was determined after the chemogenetic stimulation of Arc cells that had been TRAPed to express an activating DREADD following the injection of ghrelin (the ghrelin-TRAPed group) or in response to saline injection (henceforth referred to as the background-TRAPed group, reflecting the fact that there are invariably some background *Fos*-expressing cells in this area in the control situation). Excitation was induced by the delivery of the DREADD agonist CNO at two different concentrations (0.3 and 1 mg/kg), with controls receiving the vehicle (see Section 4.3) in a cross-over design (Figure 1A). Indeed, we were able to visually identify the location of DREADD expression, which appeared to be confined to the Arc (Figure 1A).

Statistical analyses revealed a general effect of the treatment (F_[2,51]_ = 69.733, *p* < 0.001), as well as a time × treatment interaction (F_[8,51]_ = 10.179, *p* < 0.001) on food intake. Specifically, although both the low and the high dose of CNO increased food intake compared to the vehicle at all time points (0.3 mg/kg CNO vs. vehicle: 0.5 h, *p* = 0.003; 1.5 h, *p* < 0.001; 3 h, *p* < 0.001; 6 h, *p* < 0.001; and 24 h, *p* = 0.007; 1 mg/kg CNO vs. vehicle: 0.5 h, *p* < 0.001; 1.5 h, *p* < 0.001; 3 h, *p* < 0.001; 6 h, *p* < 0.001; and 24 h, *p* < 0.001), food intake was greater in the higher versus the lower CNO dose (0.3 mg/kg CNO vs. 1 mg/kg CNO: 3 h, *p* = 0.009; 6 h, *p* < 0.001; and 24 h, *p* < 0.001) (Figure 1B). No general effect of sex was found, and thus males and females were pooled together for analyses.

To help evaluate the dynamics of the feeding response following the chemogenetic stimulation of ghrelin-activated neurons relative to that induced by ghrelin, we included data from a previously published cohort [24] showing the time course of the feeding response following a single subcutaneous (s.c.) ghrelin injection. New analyses of this dataset showed a general effect of treatment on food intake (F_[1,8]_ = 21.099, *p* = 0.002). Specifically, ghrelin increased food intake at 1 h (*p* = 0.029) and 3 h (*p* = 0.003) post-injection, while a trend was observed after 24 h (*p* = 0.089) (Figure 1C).

### 2.2. CNO-Induced Food Intake Is Greater in the Ghrelin-TRAPed Group Compared to the Background-TRAPed Control Group

An additional analysis of the 3 h time point was performed to compare the effects of TRAPing with either ghrelin or saline on food intake. In accordance with our first analysis, the treatment increased food intake (F_[1,20]_ = 132.509, *p* < 0.001). We likewise found a treatment × TRAP group interaction (F_[1,20]_ = 11.329, *p* = 0.004). Although background *Fos* expression (i.e., background-TRAPed group injected with CNO) increased food intake per se (background-TRAP + CNO vs. background-TRAP + vehicle, *p* < 0.001), this hyperphagic effect was larger when mice were initially treated with ghrelin (ghrelin-TRAP + CNO vs. background-TRAP + CNO, *p* = 0.011) (Figure 2A). No general effect of sex was found, and thus males and females were pooled together for these analyses.

### 2.3. Visualization of Ghrelin-Activated Cells in the Arc in TRAP2 Mice

Here we used the TRAP system to visualize the cells activated by peripheral ghrelin administration. To this end, *TRAP2* mice on a tdTomato reporter background (*TRAP2:Ai14* mice) received either ghrelin or saline during TRAPing. The distribution of cells activated/TRAPed in the Arc after ghrelin administration was similar to that observed previously in mice treated with exogenous ghrelin [30] (Figure 2B), and more cells were TRAPed in the Arc in the ghrelin-TRAPed group compared to the background-TRAPed group (t_[7]_ = 2.819, *p* = 0.026; Figure 2C).

### 2.4. Chemogenetic Stimulation of TRAPed Arc Cells Impacts on Food Choice in a Sex-Dependent Manner

Since the ghrelin signaling system has been implicated in food-seeking and food choice behavior [11,12], we exposed the mice first to a 15 min buried chocolate test (in which latency to treat discovery and subsequent chocolate intake were measured) and then gave them free access to chocolate and chow for 1 h. Searching for a buried chocolate is linked to motivation, after which we could then explore food choice. Since most of the statistical tests showed a sex × treatment interaction, the dataset was split according to sex for further analyses.

Regarding chocolate intake, the statistical analysis revealed an effect of the treatment to increase acute 15 min chocolate consumption during the behavioral test (F_[1,20]_ = 10.741, *p* = 0.004), in which the ghrelin-TRAPed female, but not the male, group that received CNO consumed more chocolate compared to their control peers (ghrelin-TRAP + CNO vs. ghrelin-TRAP + vehicle, *p* = 0.008) (Figure 3A). Although chocolate intake during the 1 h choice period did not differ between groups, CNO-treated, ghrelin-TRAPed females consumed a greater amount of chocolate compared to their control counterparts overall (i.e., 15 min behavioral test and 1 h food choice paradigm) (treatment: F_[1,20]_ = 47.277, *p* < 0.001; treatment × TRAP group: F_[1,20]_ = 6.634, *p* = 0.020; treatment × sex: F_[1,20]_ = 15.651, *p* = 0.025; post hoc in females: ghrelin-TRAP + CNO vs. ghrelin-TRAP + vehicle, *p* = 0.006) (Figure 3A). During the 1 h free-access period to both chocolate and chow, only the CNO-treated, ghrelin-TRAPed female group consumed more calories overall compared to their respective controls (treatment: F_[1,20]_ = 12.420, *p* = 0.003; treatment × sex: F_[1,20]_ = 6.343, *p* = 0.022; post hoc in females: ghrelin-TRAP + CNO vs. ghrelin-TRAP + vehicle *p* = 0.010). Interestingly, besides consuming more chocolate, CNO-injected, ghrelin-TRAPed female mice consumed likewise more calories from chow than both their control vehicle-injected (ghrelin-TRAP + CNO vs. ghrelin-TRAP + vehicle: *p* < 0.001) and CNO-treated, background-TRAPed counterparts (ghrelin-TRAP + CNO vs. background-TRAP + CNO: *p* = 0.038) (treatment: F_[1,20]_ = 16.189, *p* < 0.001; treatment × TRAP group: F_[1,20]_ = 4.647, *p* = 0.046; treatment × sex: F_[1,20]_ = 3.367, *p* = 0.084; general effect of TRAP group F_[1,20]_ = 4.493, *p* = 0.049) (Figure 3A). Total caloric intake was, as expected, influenced by the treatment (F_[1,20]_ = 51.384, *p* < 0.001) and interaction effects were found (treatment × TRAP group: F_[1,20]_ = 9.622, *p* = 0.006; treatment × sex: F_[1,20]_ = 14.561, *p* = 0.001). Total caloric intake was increased in both male and female CNO-treated mice (males: ghrelin-TRAP + CNO vs. ghrelin-TRAP + vehicle, *p* = 0.025; females: ghrelin-TRAP + CNO vs. ghrelin-TRAP + vehicle, *p* < 0.001; background-TRAP + CNO vs. background-TRAP + vehicle, *p* = 0.028) (Figure 3A). Neither the treatment (CNO or vehicle) nor the TRAP group (ghrelin or background) affected the latency to find the buried chocolate (Figure 3B).

### 2.5. AgRP Neurons Are Targeted in Both the Ghrelin-TRAPed and Background-TRAPed Groups

To help find an explanation for the increased food intake in the control group (background-TRAP), we investigated whether in the no-TRAP, background-TRAPed, and ghrelin-TRAPed groups the activated cells included the orexigenic AgRP neurons (a key population driving food intake in the Arc [31]). Since AgRP can be difficult to assess in colocalization studies with the FOS protein via immunohistochemistry (as it requires colchicine treatment, a known cause of *Fos* expression), we utilized the RNAscope technique. We found that, in both background-TRAPed and ghrelin-TRAPed groups, AgRP neurons were activated upon CNO administration (i.e., expressed *Fos*). Indeed, the percentage of AgRP cells expressing *Fos* upon CNO (1 mg/kg) in the ghrelin-TRAPed group (40.0% ± 8.7; *n* = 8), background-TRAPed group (22.4% ± 9.7; *n* = 4), and the group in which no TRAPing took place (7.0 ± 2.1%; *n* = 2) did not differ significantly (F^[2,11]^ = 2.520, *p* = 0.135; Figure 4A). However, albeit not reaching significance, the average percentage of *Fos*-expressing AgRP cells was almost twice as high in the ghrelin-TRAPed group compared to the background-TRAPed group (40.0% and 22.4%, respectively). Notably, in line with this, the linear regression analysis confirmed that the higher the percentage of AgRP cells expressing *Fos*, the higher the food intake was (i.e., 3 h after 1 mg/kg CNO administration; r = 0.576, *p* = 0.050; Figure 4B). Figure 4C illustrates the representative fluorescence images of RNAscope for *Agrp* and *Fos*.

## 3. Discussion

In the present study, we provide proof-of-concept that the Fos-TRAP system can serve as an alternative and complementary strategy for the functional mapping of hormone-sensitive cells that underlie the central hormone effects on behavior. By means of *TRAP2* mice, we gained genetic access to the Arc cells expressing *Fos* in response to the peripheral injection of ghrelin, allowing these populations to be visualized and chemogenetically controlled to explore their function. The number of Arc cells TRAPed and activated was greater in the ghrelin-TRAPed group than in the control background-TRAPed group. In line with this, the chemogenetic stimulation of TRAPed Arc cells increased food intake and altered food choice (in favor of chow rather than chocolate), most notably in the ghrelin-TRAPed group. Interestingly, the CNO-induced feeding response likewise correlated with a higher CNO-induced activation of AgRP neurons.

The distribution of Arc cells activated by ghrelin (here achieved through TRAPing *Fos* expressing cells in *TRAP2:Ai14* mice) resembled the distribution pattern of previous studies in which regular FOS immunohistochemistry was used [31]. Additionally, in accordance with such studies, we noted background activation even in the control situation in which TRAPing occurred after saline injection and at a time point when FOS expression ought to be lowest in the Arc (i.e., around 6 h into the light phase) [32]. In fact, we know very little about these background-activated cells, including their identity and whether they form part of the orexigenic neurocircuitry.

To TRAP ghrelin-activated cells, we used a dose of ghrelin previously shown to induce Fos in the Arc and to induce an orexigenic response [24]. Subsequent chemogenetic stimulation of these ghrelin-TRAPed cells via CNO induced an orexigenic response that depended on the dose of CNO and appeared to be similar to that induced by ghrelin administration in wild-type mice. Although the feeding response to CNO was greater in ghrelin-TRAPed than in background-TRAPed mice, the latter also increased their food intake, highlighting the fact that background-activated cells must form part of the orexigenic neurocircuitry. These data identifying the background-TRAPed Arc cells as orexigenic are important since they are relevant for all studies employing saline injection as a control group. Since AgRP neurons in the Arc play a pivotal role in stimulating food intake [33] and form part of the central ghrelin signaling system [34], we used RNAscope to explore the extent to which AgRP neurons become active (i.e., express *Fos*) upon CNO administration in both ghrelin-TRAPed and background-TRAPed groups. Importantly, we here uncovered that a fifth of these background-activated cells express AgRP. Consistent with the food intake data, however, we found that twice as many orexigenic AgRP neurons were activated by CNO in the ghrelin-TRAPed group than in the background-TRAPed group and that the number of AgRP neurons activated correlated with the food intake response.

Besides an increase in food intake, the chemogenetic stimulation of ghrelin-activated cells also impacted on food choice (although only in female mice) without altering chocolate-seeking behavior. When subsequently given the choice between regular chow and chocolate, females in the ghrelin-TRAPed group increased their chow intake, while those in the background-TRAPed group did not. These data resonate with previous studies in rats showing that central ghrelin administration favors the consumption of a healthier standard grain-based chow over energy-dense foods, such as sugar, lard or a high fat diet in a food choice paradigm [11,12,19]. Our results likewise indicate that ghrelin-activated circuits in the Arc may contribute to food choice, which is in line with a previous study showing that AgRP administration into the paraventricular hypothalamus, a downstream target of (ghrelin-targeted) AgRP cells in the Arc [35], caused rats to redirect their food preference towards an enhanced intake of a healthier chow.

Interestingly, we found this effect on food choice to be sex dependent. Indeed, only females in the ghrelin-TRAPed group increased their chow intake while having access to both chocolate and chow. Given the paucity of currently available data, more basic research in animal models is needed to elucidate the sexually dimorphic feeding behaviors and the pathways involved. Notably, however, there are studies in humans that support the existence of sex-dependent differences in terms of food preference and food craving, regarding the kind of food craved, the intensity of the craving and how the latter is regulated [36].

Our results confirm that it is important to dissect which pathways and brain areas are involved in different aspects of feeding behavior, and how they are involved. The TRAP approach used here proved to be effective at exploring the role of a hormone in a specific brain area (in our case ghrelin in the Arc) regarding feeding behavior. In the long run, knowledge about the role(s) of certain brain areas in feeding behavior can lead to the discovery of treatable targets for which drugs may be developed that support psychologic or surgical treatments for eating disorders.

There are, however, several limitations in using the Fos-TRAP system to explore hormone-responsive brain circuits as we do here for ghrelin. For example, although in our case, both GHSR and the DREADD are G protein-coupled receptors and their activation engages overlapping intracellular transduction pathways (i.e., engaging Gq-proteins) [37,38], the receptors targeted by another potential hormone of interest would not necessarily engage similar signaling transduction pathways as the DREADD would. Therefore, in such cases, the DREADD-mediated activation of these hypothetical cells may not faithfully reflect the physiological condition. On the other hand, it is important to consider that the hormone of interest may penetrate (and thus target) additional areas in the brain other than those situated within the boundary of the injection site, which could result in different outcomes. Additionally, in the case of ghrelin, its administration to *TRAP2* mice would not only activate GHSR-responsive cells, but also cells in the virus-injected area within the Arc that would be activated downstream of the GHSR cells, thus mimicking the activation of an ensemble of ghrelin-activated cells. Finally, another important issue we encountered was that background-TRAPed cells may be engaged in the responses that we examined. Despite all these limitations, *TRAP2* mice are a versatile tool to access and take control over hormone-response neural circuits, in particular when hormone-receptor Cre mice are not available or are different to wild-type mice due to the insertion effects in the receptor locus (as is the case for *Ghsr-IRES-Cre* mice).

Our study highlights the opportunities of using the Fos-TRAP system to control and manipulate the populations of neurons controlling feeding behaviors, here exemplified by targeting the ghrelin-responsive population in the Arc that drive food intake and contribute to food choice. We further highlight the importance of monitoring background-activated cells in control mice when using the Fos-TRAP system, since such cells could be contributing to the response observed. In a nutshell, activating ghrelin-TRAPed Arc neurons is orexigenic and this effect can be discriminated from background activation based on the magnitude of the response.

## 4. Materials and Methods

### 4.1. Mice

All studies were carried out on adult male and female double-transgenic *TRAP2:Ai14* mice, which were derived from crosses between homozygous *TRAP2* mice (Fos^tm2.1(icre/ERT2)Luo^/J; The Jackson Laboratory, Bar Harbor, ME, USA; #030323) and homozygous *Ai14* tdTomato reporter mice (B6.Cg-Gt(ROSA)26Sor^tm14(CAG-tdTomato)Hze^/J; The Jackson Laboratory; #007914). The genotyping of the offspring (heterozygous for both transgenes) was performed as described previously [29]. After weaning, mice were group housed (2–5 mice per cage) and left undisturbed until the start of the experimental procedures.

Mice were kept at 20–22 °C and 50% humidity on a 12 h dark–light cycle (lights on at 7:00 a.m.). Unless otherwise stated, mice had ad libitum access to water and standard chow (2016 Teklad diet; Envigo, Cambridgeshire, UK; 3.0 kcal/g).

All experiments were approved by the local Ethics Committee for animal care in Gothenburg, Sweden (Göteborgs djurförsöksetiska nämnd; permit numbers 132-2016, approved 25 January 2017, and 3112-2020, approved 26 August 2020) and complied with European guidelines (Decree 86/609/EEC).

### 4.2. Stereotaxic Surgery and Confirmation of DREADD Expression

To chemogenetically target Arc cells, the first step involved injecting a viral vector that Cre-dependently expressed a DREADD receptor into the Arc in *TRAP2:Ai14* mice via stereotaxis. To this end, male (*n* = 14) and female (*n* = 9) *TRAP2:Ai14* mice (2 to 3 months old; 27.1 ± 0.9 g body weight) were given an injection of the analgesic Rimadyl^®^ (5 mg/kg; Orion Pharma Animal Health, Sollentuna, Sweden) and then anesthetized with isoflurane prior to being placed in a stereotaxic frame. After the exposure of the skull, a local anesthetic was applied (Xylocaine 10%; AstraZeneca, Cambridge, UK). Guided by the stereotaxic coordinates (see below), two holes were drilled in the skull for the subsequent bilateral delivery of the AAV8 viral vector pAAV-hSyn-DIO-HA-hM3D(Gq)-IRES-mCitrine (0.3 µL, 2.3 × 10^13^ particles/mL, 0.2 µL/min; provided by Bryan Roth, viral prep #50454; http://n2t.net/addgene:50454 (accessed on 1 December 2021); RRID:Addgene_50454; Addgene, Watertown, NY, USA) using a 31 gauge stainless steel needle connected via vinyl tubing to a Hamilton syringe placed in an infusion pump. The injection volume was optimized prior to the study in order to minimize the spreading of the viral vector outside the Arc. The following coordinates were used for the injection of the vector into the Arc: 1.05 mm posterior to bregma, 1.25 mm lateral to the midline and 5.9 mm ventral of the skull surface at bregma. After injection, the needle was kept in place for an additional 10 min period and then slowly retracted to ensure full diffusion from the needle tip. After surgery, mice were single housed and allowed to recover for at least one week.

The correct placement of the needle tip from the viral injections was confirmed post-mortem. Mice were deeply anaesthetized with a mixture of Sedastart vet.^®^ (1 mg/kg; Produlab Pharma B.V., Raamsdonksveer, The Netherlands) and Ketalar^®^ (75 mg/kg; Pfizer AB, New York, NY, USA), prior to being perfused transcardially with heparinized 0.9% saline followed by 4% paraformaldehyde (PFA) in a 0.1 M phosphate buffer (PB). The brains were dissected, post-fixed overnight at 4 °C in 4% PFA solution and cryoprotected in 0.1 M PB containing 25% sucrose at 4°C until cryosection. The coronal sections containing the Arc (30 μm) were then cut using a cryostat and stored in tissue storage solution (25% glycerin, 25% ethylene glycol, 50% sterile 0.1 M PB) at −20 °C until further processing.

Free-floating, Arc-containing sections were processed for the immunohistochemical detection of the DREADD-fused human influenza hemagglutinin (HA) tag in order to confirm the correct placement and subsequent DREADD expression in all groups. The sections were permeabilized for 1 h in 0.1 M phosphate-buffered saline (PBS) + 0.5% Triton X-100 and then incubated for an additional 1 h in blocking buffer (0.1 M PBS + 0.5% Triton X-100 + 1% bovine serum albumin + 5% normal goat serum). For these first two steps, sections were kept at room temperature on a plate shaker at moderate speed. Finally, they were incubated with a rabbit anti-HA antibody in a blocking buffer (1:500; #3724S; Cell Signalling Technology, Danvers, MA, USA) overnight, rocking, at 4 °C. Sections were afterwards rinsed (3 × 10 min, rocking, at room temperature in 0.1 M PBS + 0.5% Triton X-100) prior to being incubated with an Alexa Fluor 488 goat anti-rabbit secondary antibody in a blocking buffer (1:250; #A11008; Invitrogen, Carlsbad, CA, USA) for 2 h, rocking, at room temperature. Sections were rinsed, counter stained with DAPI (1:5000 in 0.1 M PBS + 0.5% Triton X-100), rinsed again and mounted onto glass slides and coverslipped with ProLong Diamond Antifade mounting medium (#P36970; Thermo Fisher Scientific, Waltham, MA, USA). The slides were stored at 4 °C until image acquisition. Images of the Arc (proximate to the injection site) were captured using a DMRB fluorescence microscope (20X/N.A. 0.50; Leica Microsystems, Wetzlar, Germany).

### 4.3. TRAP Induction

The Fos-TRAP technique [28,29] was used to gain genetic access to ghrelin-activated Arc cells, including TRAPing neurons expressing *Fos* in response to ghrelin/vehicle injection during a time window defined by the delivery of 4-OHT. To limit background *Fos* expression (i.e., neurons that also have the potential to be TRAPed), mice were handled regularly over one week and were also habituated to the injection procedure (with s.c. saline injections) on three consecutive days. Three weeks after surgery (by which time the viral vector is expected to be expressed), mice were divided into three groups: a background-TRAPed group (*n* = 8: males, *n* = 4; females, *n* = 4) together with a no-TRAP group (males, *n* = 2), both receiving an s.c. injection of saline, and a ghrelin-TRAPed group (*n* = 13: males, *n* = 8; females, *n* = 5), that received a 3 mg/kg body weight s.c. injection of ghrelin instead (dissolved in saline; #1465; Tocris, Bristol, UK). The injections were performed during the light phase (11:00 a.m.) and groups were counterbalanced with respect to time of the day. Three hours later, ghrelin-TRAPed and background-TRAPed groups were injected with 4-OHT (#H6278-50MG; Sigma-Aldrich, Schnelldorf, Germany), which was dissolved in an aqueous solution (saline, 2% tween 80, 5% DMSO, and 2 mg/mL 4-OHT) and administered intraperitoneally (i.p.) (25 mg/kg). The no-TRAP group received a vehicle instead, which consisted of a mixture of saline, 2% tween 80, and 5% DMSO. We waited for another two weeks for the DREADDs to be expressed before commencing the experiments.

### 4.4. CNO Administration and Food Intake Measurements

To activate TRAPed Arc cells, male and female mice were i.p. injected with two different doses of CNO (0.3 and 1 mg/kg body weight; #4936/10; Bio-Techne Ltd., Abingdon, UK). CNO was dissolved in a vehicle (0.3 mg/kg CNO: saline, 0.3% DMSO, and 0.003 mg/mL CNO; 1 mg/kg CNO: saline, 1% DMSO, and 0.01 mg/mL CNO). Injections were performed in a cross-over fashion: every animal at one point received either CNO or an equal volume of vehicle, always with at least one wash-out day in between. CNO or saline were administered and food intake was manually measured at 0.5, 1.5, 3, 6 and 24 h post-treatment using calibrated scales that had a precision of 1 mg. Experimental groups were as follows: background-TRAP + vehicle (*n* = 8: males, *n* = 4; females, *n* = 4), background-TRAP + CNO (*n* = 8: males, *n* = 4; females, *n* = 4), ghrelin-TRAP + vehicle (*n* = 13: males, *n* = 8; females, *n* = 8) and ghrelin-TRAP + CNO (*n* = 13: males, *n* = 8; females, *n* = 5).

Food intake data available in our lab from a previously published cohort of ghrelin-injected wild-type adult male mice (*n* = 9) [24] was re-analyzed for qualitative comparison of s.c. ghrelin-induced feeding response to that observed in the *TRAP2* mice after the chemogenetic stimulation of ghrelin-activated cells in the Arc. The food intake data was analyzed in g instead of g/kg body weight to allow direct comparison, and the 1 h food intake time point was added (of note: this time point was not included in the original publication). Briefly, food intake measurements (1, 3 and 24 h) were conducted following an s.c. injection of ghrelin (3 mg/kg body weight) in a cross-over fashion.

### 4.5. Visualization of Ghrelin-Activated Cells in TRAP2 Mice

For the visualization of TRAPed cells, and additional cohort of 4 to 5 months old double-transgenic female *TRAP2:Ai14* mice (*n* = 9) received the same stimulus for TRAPing as described above (i.e., ghrelin (*n* = 5) or saline (*n* = 4)), but 4-OHT was administered i.p. at the same time as ghrelin or saline, since it was dissolved in a vehicle with different kinetics [39], specifically in Chen Oil (four parts sunflower seed oil (#S5007-250ML; Sigma-Aldrich) and one part castor oil (#259853-250ML; Sigma-Aldrich)). The 4-OHT was dissolved in Chen Oil similar to as described before [29]. Briefly, 4-OHT was dissolved to a final concentration of 20 mg/mL in ethanol at 37 °C for 15 min. For use, the dissolved 4-OHT was mixed with the same volume of Chen Oil, and ethanol was thereafter evaporated via vacuum centrifugation. The final concentration of 4-OHT was injected i.p. on the same day (dose: 50 mg/kg).

One week after TRAPing the cells for visualization, mice were deeply anesthetized with the mixture of drugs described above, prior to being perfused transcardially following the above-mentioned protocol. The brains were harvested and cut, and the 28-μm thick sections were stored as reported previously [40]. All the sections were counterstained with DAPI, coverslipped with ProLong^®^ Diamond Antifade mountant and stored in the dark at 4 °C until imaging.

Images for the quantification of the TRAPed cells were captured using a laser scanning confocal microscope (LSM 700 inverted, Zeiss, Oberkochen, Germany) equipped with a Plan-Apochromat 20×/0.8 air objective (used at the Centre for Cellular Imaging at Gothenburg University). Tile scans (3 × 3) and Z-stacks (optical section of 1 µm) of the Arc-containing sections were captured unilaterally (two sections per mouse at these approximate levels: 1.55 and 1.79 caudal to Bregma). The Z-stack images were processed using the maximum intensity projection function in the Zen Black software (Zeiss). The final images were then stitched and the tdTomato^+^ (i.e., TRAPed) cells manually counted in ImageJ/Fiji (NIH, Bethesda, MD, USA) using the cell counter plug-in. The mean number of TRAPed cells per hemisection (and averaged per two blind countings) was calculated, then averaged for each brain and ultimately for each experimental group.

### 4.6. Palatable Food Seeking and Food Choice

A 15 min buried chocolate test was used to assess the role of ghrelin-activated Arc cells on seeking for a highly palatable treat (milk chocolate; Marabou, Mondélez International, Upplands Väsby, Sweden; 5.5 kcal/g) after CNO/vehicle injection. Several days prior to testing, mice were given a small piece of chocolate (about 0.5 g) for habituation to its taste. All mice were injected with either CNO (1 mg/kg) or vehicle one day apart and in a cross-over fashion. The behavioral test started 30 min post-injection of the CNO/vehicle. Briefly, each mouse was momentarily placed in an ancillary cage while a piece of pre-weighed chocolate (4 g) was hidden under the bedding in the animal’s home cage, at a random location on each occasion. Each mouse was then placed back into the home cage and allowed to freely explore the space without interruption for 15 min. The behavioral test was recorded using a video camera. The latency to find the buried treat and the subsequent amount of chocolate consumed was measured at the end of the session. Chow access was withheld during the test.

Following the buried chocolate test, pre-weighed regular chow was re-introduced in the home cage and mice were offered free access to both chocolate and chow for 1 h. Food choice was estimated by measuring the intake of each dietary component.

### 4.7. Fluorescent In Situ Hybridization Using RNAscope

Fluorescent in situ hybridization using RNAscope was performed to study the extent to which CNO-activated cells (i.e., expressing FOS protein; *Fos* probed) co-expressed AgRP (*Agrp* probed) in the Arc of both background- and ghrelin-TRAPed groups. Mice were deeply anesthetized with the same mixture described above, prior to being perfused transcardially 90 min post-CNO injection (10 am–1 pm), and groups were counterbalanced with respect to the time of day. The brains were harvested, cut and the 14 μm thick sections stored as previously described [40]. The *Agrp* probe (#400711-C2; Advanced Cell Diagnostics, Hayward, CA, USA) contained 16 oligonucleotide pairs and targeted region 11–764 (Acc. No. NM_001271806.1) of the *Agrp* transcript. The *Fos* probe (#316921 Advanced Cell Diagnostics) contained 20 oligonucleotide pairs and targeted region 407–1427 (Acc. No. NM_010234.2) of the *Fos* transcript. The protocol used was identical to that thoroughly described previously [40]. The *Agrp* probe was labelled with Cy5 (1:2000; #NEL745001KT; Akoya Biosciences, Menlo Park, CA, USA), and the *Fos* probe with opal520 (1:500; FP1487A; PerkinElmer, Waltham, MA, USA). All the sections were counterstained with DAPI, coverslipped with ProLong^®^ Diamond Antifade mountant and stored in the dark at 4 °C until imaging.

Images for the quantification of the RNAscope data were captured using a laser scanning confocal microscope (LSM 700 inverted, Zeiss) equipped with a Plan-Apochromat 40x/1.3 Oil DIC objective (used at the Centre for Cellular Imaging at Gothenburg University). Tile scans (3 × 3) and Z-stacks (optical section of 1 µm) of the Arc-containing sections were captured unilaterally, with two sections per animal close to the injection site. Laser intensities for the different channels were kept constant throughout the imaging process. The Z-stack images were processed using the maximum intensity projection function in the Zen Black software (Zeiss). The final images were then stitched, the channels were merged and the cells automatically counted using QuPath software [41] (version 0.3.0). DAPI-identified cells with >3 dots/cell were defined as being positive for a given peptide. The quantification of the co-expression per hemisection was averaged for each brain and, ultimately, for each experimental group.

### 4.8. Statistics

Data were analyzed using the program IBM SPSS Statistics 27 (IBM Corp., Armonk, NY, USA). In Figure 1B, we used a two-way repeated measures (r)ANOVA (with sex and treatment (saline, 0.3 CNO or 1 mg/kg CNO) as the “between factors” and Time as the “within factor”) to assess the impact of these parameters on cumulative food intake in the ghrelin-TRAPed group. The effect of ghrelin on cumulative food intake was assessed with a rANOVA (Figure 1C; with treatment (ghrelin, saline) and Time as the “within factors”). In Figure 2C, an independent sample Student’s *t*-test was performed to compare the average number of TRAPed cells between the TRAP groups. In Figure 2A and Figure 3, comparisons were carried out with two-way rANOVAs (With TRAPing (saline or ghrelin) and sex as the “between factors” and treatment (saline or 1 mg/kg CNO) as the “within factor”). Further two-way ANOVA tests (and subsequent Tukey’s post hoc multiple comparisons analyses when applicable) were applied at each level to follow up significances and interactions of the main analysis. In Figure 4A, we used a one-way ANOVA to assess the average percentage of AgRP cells expressing *Fos* between the TRAP groups. A linear regression was performed to determine whether there was a correlation between the percentage of AgRP cells expressing *Fos* and CNO-induced food intake (Figure 4B). In case there was no effect of sex, mice of both sexes were pooled for further analyses.

Data are expressed as mean ± standard error of the mean. Statistical significance was set at *p* < 0.05, and values 0.05 ≤ *p* < 0.1 were considered to be evidence of statistical trends. Statistical annotations of the main analysis include the *p* value and its corresponding F or t ratio together with the degrees of freedom.

## 5. Conclusions

For hormone-targeted systems in the brain that express *Fos* when activated, it is possible to use the Fos-TRAP system to visualize them and explore their function through chemogenetics. We hereby demonstrate a proof-of-concept that ghrelin-activated cells in the hypothalamic Arc can be TRAPed, and thus visualized, identified and controlled, thereby allowing the dissection of their orexigenic function.

## Figures and Tables

**Figure 1 ijms-23-00559-f001:**
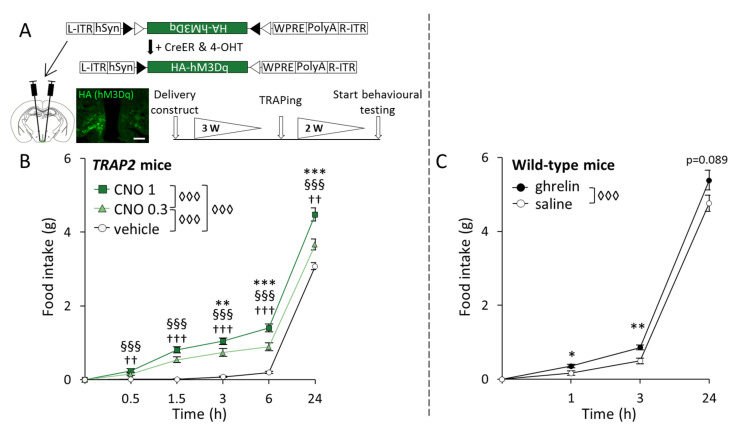
Food intake increases upon both the chemogenetic activation of ghrelin-responsive cells in the arcuate nucleus (Arc) and ghrelin injection. (**A**) The upper schematic shows the construct used and its recombination under the control of inducible Cre recombinase (CreER) and 4-hydroxy tamoxifen (4-OHT). Below is the schematic of the experimental timeline for the chemogenetic excitation of ghrelin-activated cells in the Arc, with time intervals in weeks (W). The fluorescence image is a representation of hM3Dq expression in the Arc of ghrelin-TRAPed mice, detected through anti-human influenza hemagglutinin (HA) immunohistochemistry. Scale bar: 100 µm; (**B**) Cumulative chow intake at different time points in mice that were ghrelin TRAPed (*n* = 13) and injected with 1 mg/kg CNO, 0.3 mg/kg CNO, or a vehicle in a cross-over fashion. Symbols represent: ◊◊◊ *p* < 0.001 (general effect of treatment); §§§ *p* < 0.001 (1 mg/kg CNO vs. vehicle); †† *p* < 0.01, ††† *p* < 0.001 (0.3 mg/kg CNO vs. vehicle); ** *p* < 0.01, *** *p* < 0.001 (1 mg/kg CNO vs. 0.3 mg/kg CNO); (C) Cumulative chow intake after saline or ghrelin administration to wild-type male mice (*n* = 9) in a cross-over fashion. Symbols represent: ◊◊◊ *p* < 0.001 (general effect of treatment); * *p* < 0.05 or ** *p* < 0.01 (ghrelin vs. saline). Error bars represent ± SEM in both figures.

**Figure 2 ijms-23-00559-f002:**
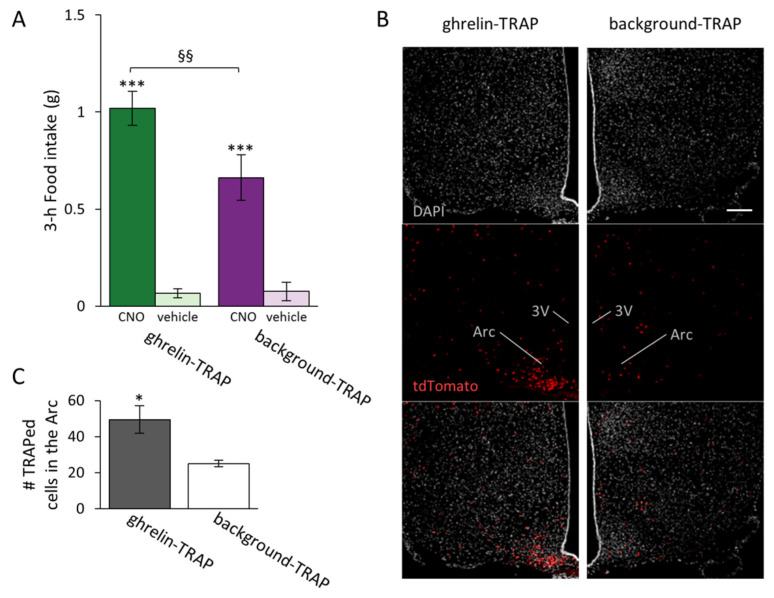
Clozapine-N-oxide (CNO)-induced food intake is greater in the ghrelin-TRAPed group than in the background-TRAPed group. (**A**) The ghrelin-TRAPed group (*n* = 13) and background-TRAPed group (*n* = 8) were injected with 1 mg/kg CNO or a vehicle in a cross-over design. Symbols represent: *** *p* < 0.001 (1 mg/kg CNO vs. vehicle); §§ *p* < 0.01 (ghrelin-TRAP + CNO vs. background-TRAP + CNO); (**B**) Representative fluorescence images of TRAPed cells (tdTomato^+^) in ghrelin-TRAPed and background-TRAPed groups. Scale bar: 100 µm, applicable to all six panels; (**C**) Number of TRAPed cells (tdTomato^+^) in the arcuate nucleus (Arc) in the ghrelin-TRAPed (*n* = 5) and background-TRAPed (*n* = 4) groups. * *p* < 0.05. Error bars represent ± SEM. Third ventricle: 3V.

**Figure 3 ijms-23-00559-f003:**
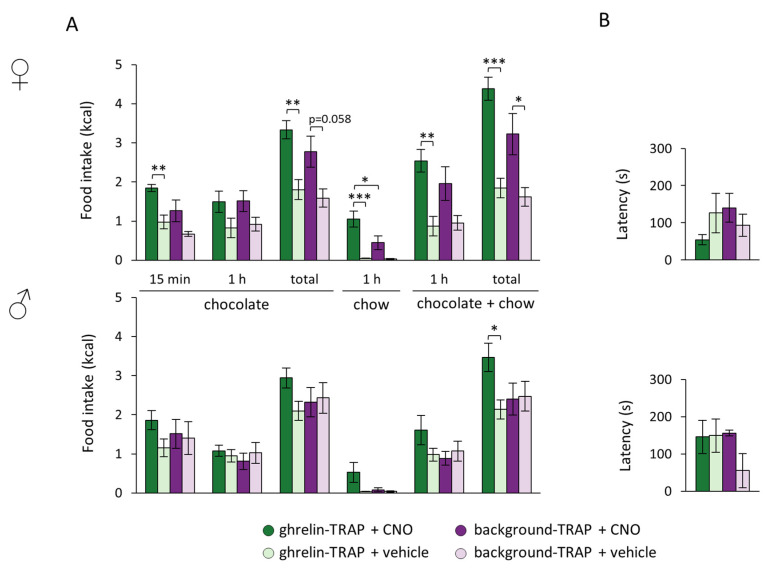
Ghrelin-responsive cells in the arcuate nucleus affected food choice in a sex-dimorphic fashion. Both the ghrelin-TRAPed (*n* = 13: males, *n* = 8; females, *n* = 5) and background-TRAPed groups (*n* = 8: males, *n* = 4; females, *n* = 4) were injected with clozapine-N-oxide (CNO) and vehicle in a cross-over design, and subjected to a chocolate search for 15 min, followed by a food choice paradigm in which they had access to both chocolate and regular chow for 1 h. (**A**) Chocolate intake (upon the 15 min behavioral test, at the end of the 1 h choice paradigm and total intake), 1 h chow intake and total energy intake (at the end of the 1 h choice paradigm and total intake) in both ghrelin-TRAPed and background-TRAPed females (upper panel) and males (lower panel). * *p* < 0.05, ** *p* < 0.01, *** *p* < 0.001; (**B**) Latency to discovering the chocolate treat in both ghrelin-TRAPed and background-TRAPed females (upper panel) and males (lower panel). Error bars represent ± SEM in all figures.

**Figure 4 ijms-23-00559-f004:**
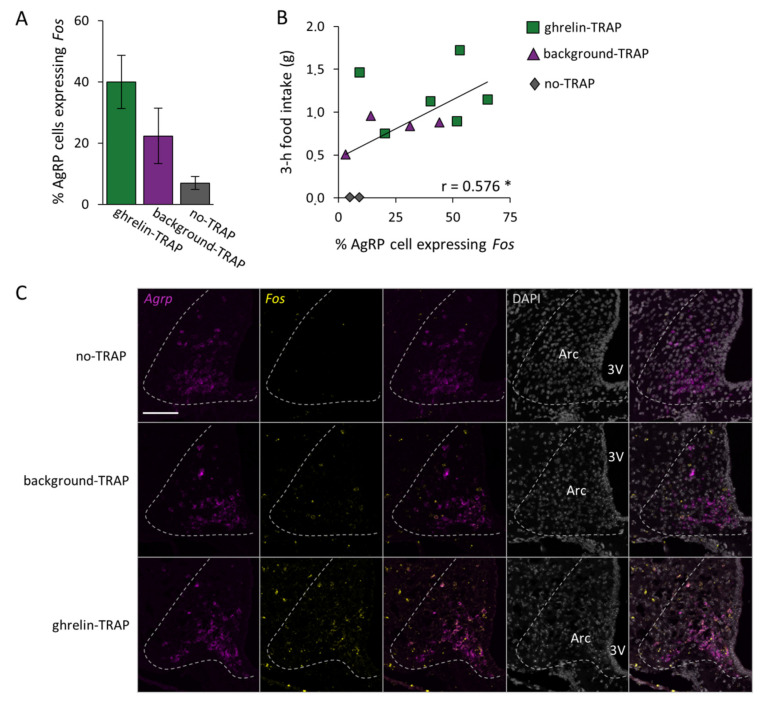
The level of AgRP activation correlates with food intake. Fluorescence in situ hybridization through RNAscope for *Agrp* and *Fos* was performed on arcuate nucleus (Arc)-containing brain sections of ghrelin-TRAPed (*n* = 6), background-TRAPed (*n* = 4), and no-TRAP (*n* = 2) mice that received clozapine-N-oxide (CNO) before sacrifice. (**A**) The percentage of *Agrp*-expressing cells that co-express *Fos* in the Arc in the three experimental groups. Error bars represent ± SEM; (**B**) The 3 h food intake time point upon CNO administration shown in relationship to the percentage of AgRP cells expressing *Fos*. The asterisk (*) indicates a significant Pearson correlation coefficient: *p* = 0.050; (**C**) Representative fluorescence images of RNAscope for *Agrp* and *Fos*, performed on no-TRAP, background-TRAPed, and ghrelin-TRAPed mice. *Agrp* is shown in magenta, *Fos* in yellow and DAPI nuclear (background) staining in gray. Scale bar: 100 µm, applicable to all 15 panels. Third ventricle: 3V.

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
