# Peer review of "TRAPing Ghrelin-Activated Circuits: A Novel Tool to Identify, Target and Control Hormone-Responsive Populations in TRAP2 Mice"

_ijms, 2022, doi:10.3390/ijms23010559_

Round 1

Reviewer 1 Report

The manuscript by Stoltenberg is focused on novel ways to target ghrelin-responsive cells by the use of TRAP mice due to severe limitations of previously published cre-lines. The authors crossed TRAP mice with a reporter strain to visualize ghrelin-responsive cells in the arcuate nucleus. By the use of viral vector strategies and chemogenetics they demonstrate similar ghrelin activation as with ghrelin administration. The authors make a point that the TRAP technology is useful for investigating effects on specific neurons in an activity dependent fashion.

This is a very well written and well performed study using advanced technologies to demonstrate useful strategies to link hormone activation with neuronal circuitry function. The manuscript is likely to interest a large number of scientists interested in neuronal function in a wide range of conditions. I have only some minor comments:

  1. Explain/write out TRAP in the abstract.
  2. It is not clear why RNAscope, although elegant, was used to demonstrate co-expression instead of immunohistochemical techniques with confocal microscopy. Could the authors please clarify.

Reviewer 2 Report

A stomach-derived hormone ghrelin targets growth hormone secretagogue receptor (Ghsr) in the brain. Ghsr-expressing neurons involve in multiple aspects of feeding-related behaviors. Cre recombinase expressing mouse models have often been used to dissect specific neuronal populations. However, due to hypomorphism resulted from the IRES-Cre insertion into the Ghsr locus, available Ghsr-IRES-Cre mouse is less ideal for functional studies of Ghsr-expressing cell population. Without generating new mouse model, authors proposed to use targeted recombination in active populations or TRAP approach to “trap” ghrelin-activated cells.

The manuscript is well written and easy to read. Although scientific novelty is limited and their discussion was superficial and repeated their results in the most parts, their observation will provide a valuable methodological option to the field. However, there are a few minor points to consider.

Experiment in figure 4 for the correlation between 3-h food intake and the population of Fos-expressing Agrp cells, does not show when mice were sacrificed after CNO injection or the feeding study.

Authors mention to use two-way repeat measures ANOVA with “sex” and treatment and TRAPing and “sex” as between factors in line 489 and 496 in Materials and Methods section, respectively. However, figure 1B and 2A are expressed as pooled data and give the impression that astatistic analysis was done with pooled data. How to handle sex in statistics needs to be corrected for precision.

In the line 71, “TRAP” indicates “targeted regulation in active populations”. However, in the original study by Luo et al. (ref. 28 and 29), “TRAP” is used the abbreviation of “targeted recombination in active populations”. Although the abbreviation is just code, it is recommended to keep consistency.

Reviewer 3 Report

The authors in this article demonstrated that neuronal TRAPing can provide a complementary / alternative technique to visualize, access and control discrete pathways, linking the action of the hormone to the function of the circuit.
I suggest algi authors to insert an abstract graphilca in this way everything is more immediate.
Do the authors think the microbiota can interfere?
The manuscript would benefit from inclusion of introducing / bridging sentences between the individual parts of the "Results" that explain the logical order and rationale for the experiments

In the Discussion, the Authors should highlight the possible clinical significance of their findings
